# An *Origanum majorana* Leaf Diet Influences Myogenin Gene Expression, Performance, and Carcass Characteristics in Lambs

**DOI:** 10.3390/ani13010014

**Published:** 2022-12-20

**Authors:** Seyed Mohammad Hadi Safaei, Mohammad Dadpasand, Mohammadreza Mohammadabadi, Hadi Atashi, Ruslana Stavetska, Nataliia Klopenko, Oleksandr Kalashnyk

**Affiliations:** 1Department of Animal Science, Faculty of Agriculture, Shiraz University, Shiraz 7194684334, Iran; 2Department of Animal Science, Faculty of Agriculture, Shahid Bahonar University of Kerman, Kerman 7616914111, Iran; 3Department of Animal Science, Bila Tserkva National Agrarian University, 09117 Soborna, Ukraine; 4Department of Animal Science, Sumy National Agrarian University, 40000 Sumy, Ukraine

**Keywords:** gene expression, growth trait, muscle, myogenin, *Origanum majorana* leaf

## Abstract

**Simple Summary:**

Nutrition affects all interactions of the body, especially the genome and the expression of genes. All organisms are always feeding, and life is not possible without nutrition. *Origanum majorana* (MO) is one nutritional additive and has many useful properties, such as antioxidant, antibacterial and antifungal properties. On the other hand, myogenin is a protein in the myogenic regulatory factor family that plays an important role in determining carcass and meat traits and is vital for the growth and development of livestock muscles. As the results of the current study show, MO might be applied in the diets of lambs in order to improve the parameters related to growth via useful reactions on myogenin gene expression.

**Abstract:**

Myogenin is a protein in the myogenic regulatory factor family that plays an important role in determining carcass and meat traits and is vital for the growth and development of livestock muscles. The objective of this study was to determine the impact of *Origanum majorana* leaf (MOL) on the myogenin gene expression of lambs. Twenty-four male Kermani lambs were used in a completely randomized design using two experimental groups (0% *Origanum majorana* L. = MOL0 and 4% *Origanum majorana* L. = MOL4). Final weight, average daily gain, hot and cold carcass weight, feed conversion ratio, empty body weight, hot and cold dressing percentage, the weight of the shoulder, loin, leg, and lean meat, and the lean/bone ratio were measured. A standard kit was used for extracting total RNA from the loin, leg, and shoulder muscles of the lambs’ tissues. The cDNA was synthesized, a real-time PCR was performed, and the obtained data were analyzed. The results of this study showed that the effect of MOL4 on dry matter intake is not significant. The MOL4 diet increased final weight by 8.22%, average daily gain by 28.57%, hot carcass weight by 11.38%, cold carcass weight by 13.43%, feed conversion ratio by 31.03%, empty body weight by 9.38%, hot dressing percentage by 2.92%, cold dressing percentage by 3.75%, shoulder weight by 56.70%, loin weight by 8.98%, leg weight by 10.90%, lean meat weight by 14.62%, and the lean/bone ratio by 2.85% (*p* < 0.05) compared to the MOL0. Along with adding MOL4 in the lambs’ diets, in comparison with MOL0, there was higher expression of myogenin in the loin (3.5 times), leg (3.9 times), and shoulder (3.6 times) muscles of the lambs. Due to the fact that adding *Origanum majorana* to the diet of the lambs enhanced the expression of the myogenin gene in the loin, leg, and shoulder muscles and increased parameters related to growth, it can be used to improve the parameters related to growth and to increase myogenin gene expression in the muscle of lambs.

## 1. Introduction

Different muscle fibers with their own specific patterns build skeletal muscle, and with their help, the organism is able to perform different movements [1]. In order to perform each of these specific roles accurately, the expression of different genes must be controlled [2]. Myogenin is a protein in the myogenic regulatory factor family that plays an important role in determining carcass and meat traits and is vital for the growth and development of livestock muscles [3,4]. Myogenin is one of the most important regulatory factors associated with myoblast differentiation [5]. This gene is located on chromosome number one in humans and mice and chromosome number five in hamsters. This gene is 2.5 kb long and contains three exons that encode a 1.5 kb mRNA [6]. The transcription factor of the basic–helixloop–helix (bHLH) protein family is encoded by myogenin. It has been shown that if the myogenin gene is knocked out in mice, they will no longer be able to form myofibers [7]. These results indicate that myogenin gene expression is very important. It has been shown that when muscles are damaged, myogenin expression begins four to five days after injury to regenerate myofibers [8]. The myogenin gene has been shown to be a vital determinant of myogenesis and has important effects on livestock and poultry meat production characteristics [9,10]. Fuso et al. [11], by studying the primary sequence of the myogenin gene, showed that CpG residues have a relatively low density. This suggests that the role of methylation may be distinct from that of classical repression mechanisms mediated by methylated CpG islands [12,13].

Phytoestrogens are plant compounds that are structurally similar to estrogen in animals and include several groups of compounds, including lignans, isoflavonoids, lactones, comets, and silicic acid residues, and are found in various plants, including cereals, peas, and fodder plants [14]. Phytoestrogens are easily broken down, not stored in tissues, and remain in the body for a short time. Therefore, when these compounds are consumed as part of a normal diet, they are likely to be safe and beneficial [15].

*Origanum majorana* (MO) belongs to the *Lamiaceae* family and is one of these plants. This plant is grown in many Mediterranean regions, including Asian, European, and North African countries [16]. MO has many useful properties, so the antioxidant, antibacterial, and antifungal properties have been reported by Aureli et al. [17], Muller et al. [18], Vera and Chane-Ming [19], and El-Ashmawy et al. [20]. According to the report by Zargary [21], thymol, myrcene, carvacrol, gamma terpinene, P-cymene, and α-pinene are the main components of MO essential oil. The most important components of MO essential oil are phenolic and alcoholic compounds. Alkaloids, terpenoids, and phenolics are the most important essential phytochemicals in medicinal plants that have antimicrobial properties, so they are used to improve animal production. A study [22] has shown that the addition of 144 and 288 ppm oregano in the diet of lambs had no effect on their daily weight gain or the feed conversion ratio (FCR). It has been shown that adding peppermint or thyme to sheep’s diet at the level of 3% dry matter increases feed intake, average daily gain, and nutrient digestibility [23]. Abdel-Moneim et al. [24] and Abdel-Wahab [25] have shown that MO can be used in the diet of broilers as a growth stimulant, and the reason has been reported as the antioxidant and antimicrobial properties of MO. In several studies, fennel has been used in sheep’s diet [15,26,27,28], and it has been shown that fennel increases muscle tissue function and increases the expression of protein delta homolog 1 (*DLK1*) and insulin-like growth factor 1 (*IGF1*) genes in different sheep tissues. The results of past studies show that MO can probably affect the expression of the myogenin gene and thereby improve some functional factors of livestock. According to the studies carried out so far, the role of feed additives, especially MO, on the expression of the myogenin gene in sheep, especially Kermani sheep, has not been studied. Thus, the aim of this study was to investigate the effect of MO feeding on the expression of the myogenin gene and dry matter intake as well as some performance and carcass parameters in growing lambs for the first time.

## 2. Materials and Methods

### 2.1. Experimental Design, Diets, and Lambs Management

To carry out this research, 24 seven-month-old male Kermani sheep with an initial weight of 26 ± 0.8 kg were used in two experimental groups (12 lambs in control and 12 lambs in treatment) in the form of a completely randomized design. These lambs were selected and bred at the research farm of Shahid Bahonar University of Kerman (Iran). The pens used to keep the lambs were 1 m × 1.3 m in size, and their floors were covered with straw. The lambs were all healthy and had no signs of disease. The lambs were treated against internal parasites using oral albendazole (Roacel), vaccinated versus enterotoxaemia (Razi Vaccine and Serum Research Institute, Karaj, Iran), and sheared. The adaptation and experimental periods were 14 and 80 days, respectively. *Origanum majorana* leaf (MOL) was obtained and dried in the shade. Samples of the experimental diets were ground (1 mm screen) and analyzed for dry matter, nitrogen (method 976.05; Kjeldahl Vap50 Gerhardt, Germany), ether extract (method 920.39; Soxhlet Model 2000 Automatic Gerhardt, Germany), and ash (method 942.05; Shimifan F-47, Tehran, Iran) according to the standard methods of AOAC [29].

Kjeldahl Vap50 Gerhardt (Germany) was used to determine crude protein content of samples according to method 976.05. To determine ash-free neutral detergent fiber (NDF) and acid detergent fiber (ADF), Van Soest et al. [30] method was performed. Metabolizable energy (Mcal/Kg DM) value of the experimental diets was calculated from the tables of Feed Specification [31]. Two diets were used: control diet (0% *Origanum majorana* leaf = MOL0) and treatment diet (4% *Origanum majorana* leaf = MOL4). Table 1 shows compositions and ingredients of MOL0 and MOL4 diets that were isonitrogenous and isocaloric including 40% forage and 60% concentrate. Nutrition was done at 08:00–09:00 a.m. and 04:00–05:00 p.m. (10% refusals) as total mixed ration (TMR). Lambs had free access to water. The daily feed was offered, and refusals were recorded to calculate the individual feed intake.

Feed intake was calculated for each lamb. Weighing the lambs was carried out at 14 day intervals before feeding in the morning. Then, the initial weight was subtracted from the final weight and the average daily gain was obtained. The ratio between average daily gain (g) and dry matter intake (kg) or feed conversion ratio was calculated. After overnight fasting at day 80 (at the end of the experiment), studied lambs were slaughtered. Then, the weight of feet, head, kidneys, bladder, spleen, lungs, heart, and liver were obtained. The digestive content and the weight of gastrointestinal tract (empty and full) were recorded. For computation of the empty body weight, the digestive content was subtracted from the final body weight at the slaughter. The weight of carcass (cold and hot) was calculated (after 24 h chilling at 4 °C). The dressing percentage (the ratio between the weight of carcass and live weight at the slaughter) was determined [32,33]. The splitting of carcass was performed longitudinally and obtained two halves based on Kashan et al. [34], and the right side was divided into 6 joints (shoulder, neck, brisket, legs, loin, and fat-tail) and then weighed independently. The lean meat (boneless and fatless meat) and bone weight were measured.

### 2.2. RNA Expression Analysis

After slaughter (day 80), samples of loin, leg, and shoulder muscle tissues were collected from each lamb. To minimize error, each tissue was sampled three times (3 biological replicates), and real-time PCR was run three times for each sample (3 technical replicates) of the three tissues (loin, leg and shoulder muscle). Therefore, the total number of samples was equal to 648 (12 × 2 × 3 × 3 × 3) samples including 2 groups of 12 lambs, 3 tissues, 3 biological replicates, and 3 technical replicates. Then, the samples were quickly placed in liquid nitrogen and then stored at −80 °C. Therefore, the total number of samples was equal to 10.

A standard kit entitled One Step RNA Reagent (Biobasic Co. Ltd., Tehran, Iran) was used for extracting total RNA (according to the manufacturer’s instructions, from 30 mg of each tissue). Extracted RNA was treated with RNase-free DNaseI to remove any contaminating genomic DNA. Then, the quality of the extracted RNA was evaluated using agarose gel electrophoresis. Not observing the DNA bands on the agarose gel and observing the 28S and 18S bands on the agarose gel confirmed the optimal quality of the extracted RNA. An oligo d(T) primer along with standard kit (#K1631, Fermentase Co., Tehran Iran) was used for synthesis of cDNA from extracted total RNA. For the myogenin target gene two primers, forward 5′-AATGAAGCCTTCGAGGCCC-3′ and reverse 5′-CGCTCTATGTACTGGATGGCG-3′ [melting temperature (Tm) = 57 °C and product size = 100 bp], and for the *GAPDH* reference gene two primers, forward 5′-ACCACTTTGGCATCGTGGAG-3′ and reverse 5′-GGGCCATCCACAGTCTTCTG-3′ [Tm = 57 °C and product size = 76 bp] were used. The final volume of each real-time PCR reaction was 15 μL, and real-time PCR was done in Rotor-Gene Q MDx device (QIAGEN Hilden, Germany). The contents of each real-time PCR reaction tube were template cDNA (1.5 µL), 2X SYBR Green PCR Master Mix (Fermentase Co., Tehran, Iran) (7.5 µL), ddH2O (4.7 µL), 10 µM forward and reverse primers (1 µL) and ROX (0.3 µL). The following program was used to perform real-time PCR reactions: 94 °C for 5 min followed by a cycle of 94 °C 20s, 57 °C 30 s, and 72 °C 30 s for 38 cycles. The Ct (cycle threshold), which is defined as the number of cycles required for the fluorescent signal to cross the threshold, was recorded. Analyzing melting curves after finishing of amplification cycles was applied to affirm that desired amplification had been done. For defining annealing temperature for studied genes (target and reference) the gradient protocol was performed. Pfaffl method [35] was employed to evaluate achieved data from real-time PCR.
$$\mathrm{ratio}=\frac{{\left(E_{\mathrm{target}}\right)}^{{\Delta \mathrm{CT}}_{\mathrm{target}(\mathrm{control}-\mathrm{sample})}}}{{\left(E_{\mathrm{ref}}\right)}^{{\Delta \mathrm{CT}}_{\mathrm{ref}(\mathrm{control}-\mathrm{sampl})}}}$$
where E_target_ is PCR yield of studied target gene, E_ref_ is PCR yield of internal control (reference) gene, and ΔCT = CT*_GADPH_* − CT_MYOG_. CT_MYOG_ is cycle threshold for target (myogenin) gene and CT*_GADPH_* is cycle threshold for control (GAPDH) gene.

### 2.3. Statistical Analysis

The mixed procedure of SAS in the format of completely randomized design was applied for data analysis [36]. The diet was fitted as fixed factor while the animal was considered a random effect in the model. The initial weight was used as the covariate for final weight, and the carcass weight was used as a covariate for analysis of carcass components.

To examine the normality of data distribution, the Pair Wise Fixed Reallocation Randomisation Test© (REST) [35] was applied. The LSD test was used to perform comparison of means (*p* < 0.05).

The below statistical model was applied to assess effect of MOL level effect and the tissue effect by the tissue × MOL interaction:X _ijm_ = μ + α_i_ + β_j_ + αβ_ij_ + ε _m(ij)_
where mean is μ, main effect of tissue at level i is α_i_, main effect of MOL at level j is β_j_, interaction effect of tissue at level i and MOL at level j is αβ_ij_, the effect of all other extraneous variables on subject m in treatment group ij is ε_m(ij)_, and dependent variable score for subject m in treatment group ij is X_ijm_.

## 3. Results

The results of this study showed that the effect of MOL on dry matter intake is not significant (Table 2). Adding MOL4 to the diet increased the final weight by 8.22%, average daily gain by 28.57%, hot carcass weight by 11.38%, cold carcass weight by 13.43%, feed conversion ratio by 31.03%, empty body weight by 9.38%, hot dressing percentage by 2.92%, cold dressing percentage by 3.75%, shoulder weight by 56.70%, loin weight by 8.98%, leg weight by 10.90%, lean meat weight by 14.62%, and the lean/bone ratio by 2.85% (*p* < 0.05) compared to MOL0.

The average cycle threshold (Ct) value of the myogenin gene in different tissues ranged from 23 to 25. The interaction between tissue and MOL feeding level was significant. Along with adding MOL4 in the lambs’ diets, in comparison with MOL0, there was a higher expression of myogenin in the loin (3.5 times), leg (3.9 times), and shoulder (3.6 times) muscles of the lambs (Table 3) (*p* < 0.05). The comparison of the expression of the myogenin gene in the loin, leg, and shoulder muscles of the lambs’ tissues at MOL4 did not show a significant difference between these studied three tissues.

The results of two-way ANOVA analysis in terms of the comparison of the means among different tissues and different levels of MO feeding for myogenin gene expression based on the LSD test is shown in Table 4, and the MOL × tissue interaction is presented in Figure 1.

## 4. Discussion

The results of the Ct values (ranged from 23 to 25) indicated that the transcript abundance of myogenin in different tissues was high [26] (Cts < 29 are strong positive reactions indicative of abundant target nucleic acid in the sample, Cts of 30–37 are positive reactions indicative of moderate amounts of target nucleic acid, Cts of 38–40 are weak reactions indicative of minimal amounts of target nucleic acid, which could represent an infection state or environmental contamination). However, factors such as instrument settings, the amount of cDNA, and the efficiency of real-time PCR affected the Ct value. Along with adding MOL4 to the diets of the lambs, compared to MOL0, there was a greater expression of myogenin in the loin, leg, and shoulder muscles of the lambs (Table 3).

Song et al. [37] and Zhao et al. [5] have demonstrated that the myogenin gene expresses in the breast muscles and the leg muscles of geese. Zhang et al. [38] studied the expression of the myogenin gene in chickens and showed that this gene expresses in the breast muscle, leg muscle, heart, liver, spleen, lung, kidney, glandular stomach, and ovarian tissues. Lv et al. [39] and Forutan et al. [40] demonstrated that the myogenin gene expresses in different skeletal muscles of sheep, such as the *soleus*, *gastrocnemius*, *longissimus dorsi,* and *extensor digitorum longus* muscles. Kuang et al. [41] reported the expression of the myogenin gene in the skeletal muscles of rabbits. In livestock and poultry, myogenin is one of the important players in myogenesis and meat production [9,10]. Some studies [3,42,43,44] reported that in early skeletal muscle satellite cells, myogenic factor 5 (*MYF5*) or myoblast determination protein (*MYOD*) is the first of the myogenic regulatory factors to be expressed while myogenin is expressed at later stages. Zhao et al. [5] showed that myogenin expression in the breast muscles and in the leg muscles of geese had a positive association with body weight and concluded that myogenin can act as a mediator of muscle growth. They reported that a high level of myogenin expression in goose indicates the important role of this gene in muscle development and differentiation. Zhang et al. [38] showed that the myogenin gene mostly expresses in the muscle of Jinghai yellow chicken, which shows the important role of this gene in muscle development and differentiation. They concluded that myogenin expression in the muscle of chicken has a positive correlation with growth traits. Masoudzadeh et al. [26] showed that enhancing the fennel seed powder level in sheep diets improves the expression of the *DLK1* gene in the femur muscle. They concluded that fennel seed powder could be used for improving animal growth and muscle mass. Another study [27] reported that adding fennel seed powder to the diets of growing sheep can increase muscle structure (mass and size of muscle fiber) by improving the *DLK1* gene expression.

Dry matter intake was not affected by adding MOL to the diets of the studied lambs (Table 2). Since moisture and NDF contents are associated with dry matter intake, this lack of change is probably due to the same contents of moisture, ADF and NDF, in the diets [45,46]. It has been shown that adding dried oregano leaves to the diets of growing lambs has no effect on dry matter intake [22]. Including oregano extract in the diets of dairy heifers reduced concentrate intake and had no effect on dry matter intake [47]. Chaves et al. [48] demonstrated that adding carvacrol to the diets of growing lambs does not affect final weight and body weight gain in the studied animals. Some studies [26,27,49] have shown that diets containing fennel seed powder increase dry matter intake. While in another study [50], including rosemary essential oils in the diet could not increase the dry matter intake in animals.

In our investigation, lambs that used MOL4 showed a higher final body weight and live weight gains in comparison to lambs fed with MOL0. This increase in daily weight and final weight obtained by the addition of 4% *Origanum majorana* leaf is probably due to the improvement of rumen fermentation in lambs. Tekippe et al. [51] showed that the presence of carvacrol in MO reduces the production of methane in the rumen. Chaves et al. [48] also demonstrated that carvacrol in MO enhances VFAs concentration in the rumen. Because VFAs are one of the primary sources of metabolizable energy in ruminants, therefore, increasing the fermentability of the diet can be very useful. In our investigation, lambs that used MOL4 showed higher FCR in comparison to lambs fed with MOL0. Since all the lambs had the same dry matter intake, but the lambs that consumed MO showed a greater daily weight gain, this increase in FCR seems reasonable. Mohiti-Asli et al. [52] concluded that including herbal additives in the diets of animals enhances FCR and growth performance and motivates the utilization of feed. Likewise, it has been shown that when fennel seed powder was included in dairy calves’ diets, FCR improved. In other studies [22,48,53], it has been shown that dried oregano leaves, carvacrol, and purslane powder supplementations in the diets of the growing lambs do not affect the FCR.

In our investigation, lambs that used MOL4 had higher hot carcass weight, cold carcass weight, and empty body weight than the lambs fed with MOL0. This is due to the higher final weight of lambs that received the MOL4 diet. Cherif et al. [54] demonstrated that the growth performance of Barbarian sheep can be increased by including nigella seeds in their diets. In some studies, it has been proven that adding plantain and chicory [55], fennel [15,26,27,56], and rosemary [57] in the diets of animals can improve carcass weight and live weight gain in comparison to controls.

In our investigation, the lambs that used MOL4 had a higher weight for the leg, loin, and shoulder than the lambs fed with MOL0. This is probably due to the higher final body weight of lambs fed with MOL4. In the same way, Karami et al. [58] showed that adding turmeric supplements to the diets of goats improved longissimus muscle and decreased back fat thickness. Dudko et al. [59] also demonstrated that including *Oreganum vulgare* and citrus supplements in the diets of growing lambs increases the depth of longissimus dorsi muscle and reduces back fat thickness. In the current study, lambs fed with MOL4 had a higher lean-to-bone ratio than the lambs fed with MOL0. This is probably due to the higher lean meat weight of lambs fed with MOL4. Furthermore, it has been shown that adding the herb–clover mixes (red and white clovers, chicory, and plantain) in the ratios improves lean to bone ratio [60].

## 5. Conclusions

The results of the current study demonstrate that MOL4 in comparison to MOL0 increases the expression of myogenin in the loin, leg, and shoulder muscles of lambs and improves the parameters related to growth via useful reactions on myogenin gene expression. Due to the fact that adding MO to the diets of lambs has enhanced the expression of the myogenin gene in the loin, leg, and shoulder muscle and increased parameters related to growth, it can be used to improve the parameters related to growth and to increase myogenin gene expression in the muscles of lambs. Since gene expression and its role in production and growth are influenced by various genetic, epigenetic, and physiological factors, for the final conclusion, it is better to consider all these factors in future research and to express this gene in more tissues and study a larger number of animals. It should be noted that the results obtained from the role of MO in the expression of the myogenin gene and its effect on growth traits open a progressive horizon for further research in this direction.

## Figures and Tables

**Figure 1 animals-13-00014-f001:**
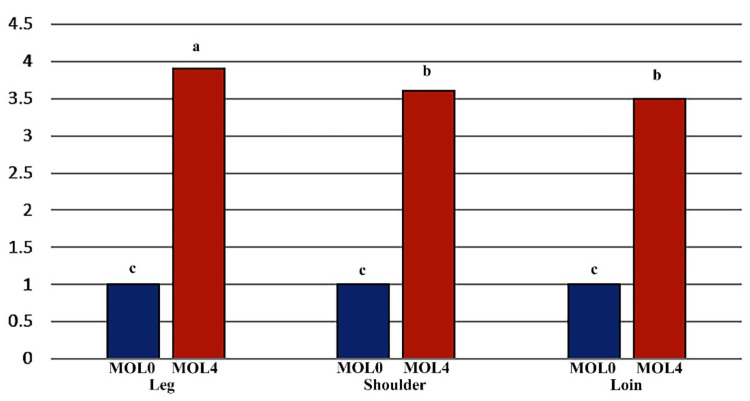
The MOL × tissue interaction for expression of myogenin in Kermani lambs in different tissues at different levels of MOL feeding. MOL0: *0% Origanum majorana* leaf, MOL4: 4% *Origanum* majorana leaf. Treatments that have at least one letter in common do not differ significantly.

**Table 1 animals-13-00014-t001:** Nutrient composition and ingredients of diets in the current study.

Ingredients	Diets *	
	MOL0	MOL4
*Origanum majorana* leaf	0	4
Chopped alfalfa hay	30	30
Ground barley grain	28	28
Chopped wheat straw	10	10
Wheat bran	13	13
Ground corn grain	9	5
Soybean meal	8	8
Vitamins (A, D and E) **	0.6	0.6
Limestone	0.3	0.3
Sodium bicarbonate	0.5	0.5
Trace-mineralized salt ***	0.6	0.6
	Chemical composition	
Organic matter (g/kg DM)	926.1	922.5
Dry matter (g/kg)	899.6	900.6
Crude protein (g/kg DM)	139.8	140
Metabolizable energy (Mcal/Kg DM)	2.52	2.52
NDFom (g/kg DM)	393.3	413.3
ADFom (g/kg DM)	240.2	240.2
Ether extract (g/kg DM)	22.5	24.3

* diets–control diet: without *Origanum majorana* leaf = MOL0, treatment diet: 4% *Origanum majorana* leaf = MOL4, ME: value of the experimental diets was calculated from the tables of Feed Specification [31], DM: dry matter, ADFom: ash-free ADF, and NDFom: ash-free NDF. ** Contains per kg; Vitamin A: 5,000,000 IU, Vitamin D: 5,000,000 IU, and Vitamin E: 500,000 IU. *** Composition: 20.5% Dynamad, 75.15% NACL, 1.025% cu-sulphate, 3.046% Mn, 0.015% EDDI-80, 0.253% Zn-sulphate, and 0.011% Na-selenide.

**Table 2 animals-13-00014-t002:** Effect of diets on dry matter intake and some performance parameters of studied lambs.

Parameters	MOL0	MOL4	SEM	*p* Value
Initial weight (kg)	26.02	26.00	0.33	0.98
Final weight (kg)	42.55	46.05	0.74	0.003
Average daily gain (g)	210	270	10.96	0.007
Dry matter intake (kg/day)	1.45	1.42	0.02	0.23
Hot carcass weight (kg)	20.38	22.70	0.38	0.02
Cold carcass weight (kg)	19.65	22.29	0.67	0.02
Feed conversion ratio * (g body weight gain/kg dry matter intake)	145	190	0.26	0.001
Empty body weight (kg)	39.10	42.77	0.64	0.004
Hot dressing percentage	47.89	49.29	0.63	0.24
Cold dressing percentage	46.65	48.40	0.56	0.15
Shoulder (kg)—percentage of right half carcass	0.97	1.52	0.12	0.02
Loin (kg)—percentage of right half carcass	1.78	1.94	0.05	0.02
Leg (kg)—percentage of right half carcass	2.66	2.95	0.07	0.004
Lean meat (kg)	13.74	15.75	0.51	0.01
Lean/bone ratio	4.20	4.32	0.30	0.001

* Feed conversion ratio: ratio of average daily gain to dry matter intake. MOL0 = control diet or 0% *Origanum majorana* leaf, MOL4 = treatment diet or 4% *Origanum majorana* leaf.

**Table 3 animals-13-00014-t003:** The effect of diets on expression of **myogenin** gene in loin, leg, and shoulder muscle of Kermani lambs.

Tissue	Relative Expression of Myogenin		SEM	*p* Value
	MOL0	MOL4		
Loin	1	3.5 *	0.13	0.03
Leg	1	3.9 *	0.14	0.01
Shoulder	1	3.6 *	0.12	0.02

Treatments marked with * have a significant difference (*p* < 0.05) in comparison to control diet (0% *Origanum majorana* leaf= MOL0) for any tissue. MOL4 = treatment diet or 4% *Origanum majorana* leaf.

**Table 4 animals-13-00014-t004:** Two-way ANOVA analysis for expression of myogenin in Kermani lambs in different tissues at different levels of MOL feeding.

Source of Variation	df	Mean Square
Tissue	2	0.065 **
MOL	1	32.000 **
Tissue × MOL	2	0.065 **
Coeff Var	3.03	

MOL: *Origanum majorana* leaf. ** shows significant differences at *p* < 0.01

## Data Availability

The data sets generated and/or analyzed during the current study are available from the corresponding author upon reasonable request.

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
