# Peer review of "An Origanum majorana Leaf Diet Influences Myogenin Gene Expression, Performance, and Carcass Characteristics in Lambs"

_animals, 2022, doi:10.3390/ani13010014_

Round 1

Reviewer 1 Report

The study entitled "Effect of Origanum majorana leaf diet on Myogenin gene expression in loin, leg and shoulder muscle of lambs" could be interesting even if it needs several clarifications in order to be published.

The title does not precisely reflect the purpose of the research.

The introduction is too long and gets lost in considerations that are not useful for the explanation of the purpose.

M&M must be completely revised. Group formation must be well formulated indicating the number of animals per group  and sex.

It is not clear from which samples the RNA was analyzed and how they were taken.

The results indicate a significance that does not correspond to that reported in the table. Some parameters analyzed are not reported in the results. 

In the statistical model the interaction is indicated and then the value of the interaction is not reported in the results.

The conclusion should be more centered on the results obtained.

The work could be interesting but too many inconsistencies make it of poor quality and not completely understandable.

Kind regards

Author Response

Thank you very much for your attention and comments.

We corrected paper Effect of Origanum majorana leaf diet on Myogenin gene expression in loin, leg and shoulder muscle of lambs based on your opinions.

We have addressed all of Reviewer’s comments as follows and in the text, they are appeared with different colors. We wish to thank the reviewer for the useful comments, which helped us to improve the quality of our manuscript.

  • The title does not precisely reflect the purpose of the research.

Effect of Origanum majorana leaf diet on Myogenin gene expression in the muscle and on dry matter intake and some performance parameters of lambs    

  • The introduction is too long and gets lost in considerations that are not useful for the explanation of the purpose.

The aerial part and especially the leaves of Origanum majorana are used all over the world as a very pleasant and aromatic spice. In addition to traditional uses, it is widely used in the treatment of gastrointestinal diseases, constipation, respiratory disorders such as asthma. It is also used as a healer and disinfectant (Tahmasebi et al., 2014).

Tahmasebi, S., Majd, A., Mehrafarin, A., Jonoubi, P. Qualitative and quantitative assessment of the essential oils of Origanum vulgare and Origanum majorana in Iran. J. Med. Plant. 2014, 2, 163–171.

In another study, Rahchamani et al. (2019) used dry licorice powder (Glycyrrhiza glabra L.) in sheep diets and showed that it increased FCR.

Rahchamani, R.; Faramarzi, M.; Moslemipor, F.; Bayat Kohsar, J. Effect of supplementing sheep diet with Glycyrrhiza glabra and Urtica dioica powder on growth performance, rumen bacterial community and some blood biochemical constituents. Iran. J. Appl. Anim. Sci. 2019, 9, 95–103.

For example, Wang et al. (2009) used rupadiar (essential oil of oregano) in sheep diet and showed that the production of ammonia nitrogen and methane in the rumen decreased compared to the control diet and volatile fatty acid (VFA) concentration increased.

Wang, C.H.; Wang, S.H.; Zhou, H. Influences of flavomycin, ropadiar, and saponin on nutrient digestibility, rumen fermentation, and methane emission from sheep. Anim. Feed Sci. Technol. 2009, 148, 157–166.

There are in different parts of the plant and seeds and depending on the cultivar, geographical location and year of growth of the plant, which after entering the gastrointestinal tract may be excreted or absorbed or broken down into compounds that are also strong phytoestrogens (Mohammadabadi et al. 2021a).

The results of the past studies show that Origanum majorana can probably affect the expression of the Myogenin gene and thereby improve some functional factors of livestock. According to the studies carried out so far, the role of food additives, especially Origanum majorana, on expression of Myogenin gene in sheep, especially Kermani sheep, has not been studied. Thus, the aim of this study was to investigate effect of Origanum majorana feeding on expression of Myogenin gene and on dry matter intake and some performance parameters in the growing lambs for the first time.

  • M&M must be completely revised. Group formation must be well formulated indicating the number of animals per group and sex.
  • It is not clear from which samples the RNA was analyzed and how they were taken.

After slaughter, samples of loin, leg and shoulder muscle tissues were collected from each sheep (3 repeats for each of the three tissues in each of the 2 groups of 12 animals, totally 216 samples), then stored at -80 °C.

 The results indicate a significance that does not correspond to that reported in the table. Some parameters analyzed are not reported in the results. .

Answer: It was corrected. Moreover, ANOVA table was added as below:

  • Table 4. Two-way ANOVA analysis for expression of myogenin in Kermani lambs in different tissues at different levels of MOL feeding.

Source of variation

df

Mean square

Tissue

2

0.065**

MOL

1

32.000**

Tissue* MOL

2

0.065**

Coeff Var

3.03

MOL: Origanum majorana leaf

  • In the statistical model the interaction is indicated and then the value of the interaction is not reported in the results.

Answer: It was done. A figure was added as below:

Figure 1. The MOL × tissue interaction for expression of myogenin in Kermani lambs in different tissues at different levels of MOL feeding. MOL0: 0% Origanum majorana leaf, MOL4: 4% Origanum majorana leaf. Treatments that have at least one letter in common do not differ significantly. 

  • The conclusion should be more centered on the results obtained.

Answer: it was done.

Answer: It was changed as below:

While, adding 4 % Origanum majorana leaf to diet increased final weight by 8.22%, average daily gain by 28.57%, warm carcass weight by 11.38%, cold carcass weight by 13.43%, conversion rate by 31.03%, empty body weight by 9.38%, warm dressing percentage by 2.92%, cold dressing percentage by 3.75%, weight of shoulder by 56.70%, weight of loin by 8.98%, weight of leg by 10.90%, weight of lean meat by 14.62%, and lean/bone ratio by 2.85% (P < 0.05), compared to the control diet (0 % Origanum majorana leaf). Along with adding 4% Origanum majorana leaf in the lambs diets, in comparison with 0% Origanum majorana leaf, there was higher expression of Myogenin in loin (3.5 times), leg (3.9 times) and shoulder (3.6 times) muscle of lambs.

Based on the results of the present study, it can be derived that Origanum majorana leaf might be applied in the diets of sheep in order to improve the parameters related to growth via useful reactions on Myogenin gene expression. Due to the fact that adding Origanum majorana to the diet of sheep has enhanced the expression of the Myogenin gene in the loin, leg and shoulder muscle and increased parameters related to growth, it can be used to improve the parameters related to growth and to increase Myogenin gene expression in the muscle of sheep.

  • Taking into account the low number of animals. Please perform the test power analysis to determine that the results obtained are reliable and do not have methodological and/or statistical errors.

Answer: It was performed.

If we refer to the published articles in the field of gene expression, we see that these numbers are not small. Considering that we had three technical replicates for each sample, the sum of the samples is

12 animals *3 tissues*3 biological replicates *3 technical replicates (control)+ 12*3*3*3 (treatment)= 324+324=648.

The power test was done for treatment group as below:

For technical replicates, we considered the average of three repeats.

                                         The SAS System           07:12 Saturday, May 1, 2010   1

                                       The POWER Procedure

                                   One-sample t Test for Mean

                                    Fixed Scenario Elements

                               Distribution                Normal

                               Method                       Exact

                               Mean                         3.666

                               Standard Deviation           0.189

                               Total Sample Size              108

                               Number of Sides                  2

                               Null Mean                        0

                               Alpha                         0.05

                                          Computed Power

                                              Power

                                              >.999

Answer: It was changed as below.

adding 4 % Origanum majorana leaf to diet increased final weight by 8.22%, average daily gain by 28.57%, warm carcass weight by 11.38%, cold carcass weight by 13.43%, conversion rate by 31.03%, empty body weight by 9.38%, warm dressing percentage by 2.92%, cold dressing percentage by 3.75%, weight of shoulder by 56.70%, weight of loin by 8.98%, weight of leg by 10.90%, weight of lean meat by 14.62%, and lean/bone ratio by 2.85% (P < 0.05), compared to the control diet (0 % Origanum majorana leaf). Along with adding 4% Origanum majorana leaf in the lambs diets, in comparison with 0% Origanum majorana leaf, there was higher expression of Myogenin in loin (3.5 times), leg (3.9 times) and shoulder (3.6 times) muscle of lambs.

Based on the results of the present study, it can be derived that Origanum majorana leaf might be applied in the diets of sheep in order to improve the parameters related to growth via useful reactions on Myogenin gene expression. Due to the fact that adding Origanum majorana to the diet of sheep has enhanced the expression of the Myogenin gene in the loin, leg and shoulder muscle and increased parameters related to growth, it can be used to improve the parameters related to growth and to increase Myogenin gene expression in the muscle of sheep.

Answer: Based on opinion of other reviewers the discussion section was changed. As we know, all dietary and environmental changes have an effect on the expression of genes and the turning on and off of genes (epigenetics), but for a detailed examination of this food additive and the way it affects genes and its mechanism of action, additional tests are needed, which in conclusionit is mentioned.

Answer: It was done.

  • Line 18: Change “On” instead of “one”.

Answer: It was done.

  • Lines 37-43: Repetitive. Rewrite it.

Answer: It was changed as below.

Along with adding 4% Origanum majorana leaf in the lambs diets, in comparison with 0% Origanum majorana leaf, there was higher expression of Myogenin in loin (3.5 times), leg (3.9 times) and shoulder (3.6 times) muscle of lambs. Based on the results of the present study, it can be derived that Origanum majorana leaf might be applied in the diets of sheep in order to improve the parameters related to growth via useful reactions on Myogenin gene expression. Due to the fact that adding Origanum majorana to the diet of sheep has enhanced the expression of the Myogenin gene in the loin, leg and shoulder muscle and increased parameters related to growth, it can be used to improve the parameters related to growth and to increase Myogenin gene expression in the muscle of sheep.

  • Line 40: Change “improve” instead of “better”.

Answer: It was done.

  • Lines 82 and 87: Please, homogenize the use of the name: Sweet marjoram, marjoram, majorana or Origanum majorana here and throughout the text.
  • Line 109: Those genes (DLK1 and IGF1) have a name, add them.
  • Line 110-113: This part is confusing and weak. Rewrite it by describing a hypothesis and a specific objective.

Answer: It was performed as below:

According to the studies carried out so far, the role of food additives, especially Origanum majorana, on expression of Myogenin gene in sheep, especially Kermani sheep, has not been studied. Thus, the aim of this study was to investigate effect of Origanum majorana feeding on expression of Myogenin gene and on dry matter intake and some performance parameters in the growing lambs for the first time.

  • Line 117: This is a low number of animals, add or describe the use of the power test of the current study.

Answer: It was performed.

If we refer to the published articles in the field of gene expression, we see that these numbers are not small. Considering that we had three technical replicates for each sample, the sum of the samples is

12 animals *3 tissues*3 biological replicates *3 technical replicates (control)+ 12*3*3*3 (treatment)= 324+324=648.

The power test was done for treatment group as below:

For technical replicates, we considered the average of three repeats.

                                         The SAS System           07:12 Saturday, May 1, 2010   1

                                       The POWER Procedure

                                   One-sample t Test for Mean

                                    Fixed Scenario Elements

                               Distribution                Normal

                               Method                       Exact

                               Mean                         3.666

                               Standard Deviation           0.189

                               Total Sample Size              108

                               Number of Sides                  2

                               Null Mean                        0

                               Alpha                         0.05

                                          Computed Power

                                              Power

                                              >.999

  • Table 1: Dear authors, are you sure that oregano replaced ground corn in the diet?
  • Table 1: If oregano replaced ground corn, how is the metabolizable energy the same between treatments?

Answer: We analyzed MOL before replacing corn with MOL. Results of analysis showed that metabolizable energy and crude protein of MOL was almost (ME=2.9 and CP=7.95) same as for corn. Hence, this very small difference in 4% becomes a very small number that can be ignored.

  • Lines 147-148: How much blood was collected? On what days was blood collected?

Answer: on day 80. But, based on opinion of other reviewer this sentences were deleted:

Blood samples were collected in tubes without clot activator from jugular (3 hours after morning feeding) at the end of the experimental period (day 80). After that, these samples centrifuged at 3000 × g for 15 min and stored for future using at -20 â—¦C. the standard kits obtained from Darmanfaraz Company (Isfahan, Iran) were used to measure cholesterol, triglycerides (TG), albumin, total protein and serum glucose. A testosterone kit obtained from Patan Gostar Eisar Company (Iran) on ELISA set (Stat Fax) was applied to determine concentration of testosterone.

  • Line 150: What was stored? Blood, serum, plasma, etc?
  • Lines 150-154: How much sample was used to each analysis?
  • Line 157: What is “DC”.

digestive content (DC)

  • Lines 154-163: At what time were meat samples collected for myogenin analysis? How much samples were collected for myogenin analysis? How were those samples stored and treated for myogenin analysis?

After slaughter (day 80), samples of loin, leg and shoulder muscle tissues were collected from each sheep (3 biological replicates and 3 technical replicates for each of the three tissues in each of the 2 groups of 12 lambs, totally 648 (12*2*3*3*3) samples). Then the samples were quickly placed in liquid nitrogen and then stored at -80 °C.

  • Line 166: How were the samples treated before DNA evaluation?
  • Line 194: Homogeneity was evaluated or only normality?
  • Line 208-209: Change “hot” instead of “warm”. Here and through the text.
  • Line 210: Does lean meat refer to a boneless meat? Please describe this in the material and methods topic.
  • Line 210: What is “cycle threshold”? Please describe this in the material and methods topic.

Answer: It was added in M&M section as below:

The Ct (cycle threshold), which defined as the number of cycles required for the fluorescent signal to cross the threshold, were recorded.

  • Lines 222-223: Repeated text on lines 213-214.
  • Lines 223-224: Why is it high? What values are the references?

(Cts < 29 are strong positive reactions indicative of abundant target nucleic acid in the sample, Cts of 30-37 are positive reactions indicative of moderate amounts of target nucleic acid, Cts of 38-40 are weak reactions indicative of minimal amounts of target nucleic acid, which could represent an infection state or environmental contamination).

  • Table 2: Are the hot and cold carcass weight measurements correct?
  • Table 2: Is the Empty body weight measurement correct?
  • Since the text contains 50 Origanum majoranas, it becomes monotonous. Therefore, I recommend that you use abbreviations.
  • In accordance with Instructions for Authors, references should be numbered in chronological order, with a number or numbers in square brackets, such as [1] or [2,3], or [4–6]. It is therefore necessary to correct all references cited in the text.
  • L73-76: Could you please provide another explanation. This sentence disrupts the flow of the text.
  • L112-113: As some factors other than gene expression must also be taken into account, this section should be written more emphatically.
  • L127: The NDF and ADF are not found by calculation. Therefore, this statement should be corrected. Please use "neutral detergent fiber (NDF) and acid detergent fiber (ADF)” instead of “NDF and ADF”, and emphasize that they do not contain ash.

To determine neutral detergent fiber (NDF) and acid detergent fiber (ADF) Van Soest et al. (1991) method was performed. NDF and ADF obtained by this method do not contain ash.

  • As part of section 2.1, please detail how organic matter, dry matter, ether extract, metabolizable energy are made or calculated.

Samples of the experimental diets were ground (1-mm screen) and analyzed for dry matter, nitrogen (method 976.05; Kjeldahl Vap50 Gerhardt, Germany), ether extract (method 920.39; Soxhlet Model 2000 Automatic Gerhardt, Germany) and ash (method 942.05; Shimifan F-47, Tehran, Iran) according to the standard methods of AOAC (2000).

Metabolizable energy (Mcal/Kg DM) value of the experimental diets was calculated from the tables of Feed Specification (National Research Council, 2007).

  • L128-132: Please write this part in a more clear and specific order. The purpose of using “10% refusals” is unclear. Do you refer to the part of the feed collected for analysis or to the 10% excess of feed that animals can consume by live weight?

The daily feed was offered and refusals were recorded to calculate the individual feed intake.

  • For clarity and to avoid repetition, I suggest that the diet that contains 0% Origanum majorana L. and 4% Origanum majorana L. be referred to as "MOL0" and "MOL4", respectively instead of the control and treatment diets.
  • In Table 1: Please use “Diets*”, “MOL0” and “MOL4” instead of “Used diets*”, “Control diet” and “Treatment diet. Their descriptions should also be corrected in the footnotes. According to the chemical composition, please provide the ingredients of the diets as grams per kilogram of dry matter.
  • L150-154: The following sentence should be removed from the text. There is no table in the text that provides information regarding cholesterol, triglycerides, albumin, total protein, and serum glucose or testosterone levels. It is important, however, that if the analysis has been performed, the results are presented in a table and discussed appropriately.
  • L155-160: References should be provided where appropriate.
  • L195: The reference section should be updated to include " REST, 2009"
  • There should also be a description of the other statistical model used for the performance and carcass parameters in section 2.3.

The diet was fitted as fixed factor while the animal was considered as random effect in the model. The initial weight was used as the co-variate for final weight, and the carcass weight was used as a co-variate for analysis of carcass components.

  • L206-219: There is a need to revamp the “Results” section, as it does not discuss the performance and carcass parameters. Each parameter examined should be included in the “Results” section.

Results of this study showed that effect of MOL4 on dry matter intake is not significant (Table 2). While, adding MOL4 to diet increased final weight by 8.22%, average daily gain by 28.57%, hot carcass weight by 11.38%, cold carcass weight by 13.43%, conversion rate by 31.03%, empty body weight by 9.38%, hot dressing percentage by 2.92%, cold dressing percentage by 3.75%, weight of shoulder by 56.70%, weight of loin by 8.98%, weight of leg by 10.90%, weight of lean meat by 14.62%, and lean/bone ratio by 2.85% (P < 0.05), compared to the MOL0.

The average cycle threshold (Ct) value of the Myogenin gene in different tissues ranged from 23 to 25. The interaction between tissue and MOL feeding level was always significant. Along with adding MOL4 in the lambs diets, in comparison with MOL0, there was higher expression of Myogenin in loin (3.5 times), leg (3.9 times) and shoulder (3.6 times) muscle of lambs (Table 3) (P<0.05). The comparison of the expression of Myogenin gene in the loin, leg and shoulder muscle of lambs tissues at MOL4 did not show significant difference between these studied three tissues.

  • L221-241: Tables presented in the discussion section should be transferred to the results section.
  • L288-291: The results of these studies should be stated.
  • L296: Please use the word “lambs” instead of “animals” where appropriate. All of the text should be considered in this manner.
  • L312-315: If it was possible to provide references to which supplements were used in which study, it would be better.
  • L288, 301, 319, 326: A sentence that begins with "in this regard" does not flow. Similar results can be achieved by substituting "in addition", "likewise", "furthermore", "in the same way" or other pertinent words in place of this phrase.
  • L329-330: There was no correlation analysis made between myogenin and working traits in the current study. Therefore, this expression may not be appropriate.
  • L330-335: Since L330-335 are similar and complementary, they are better combined and presented as one sentence.

The results of current stydy demonstrated that MOL4 in comparison to MOL0 increases expression of Myogenin in loin, leg and shoulder muscle of lambs and improves the parameters related to growth via useful reactions on Myogenin gene expression.

  • L329-343: It is important to mention not only the myogenin but also other remarkable results in the conclusion section.
  • L446-447: It is recommended that this reference be checked.
  • L3, 45, 80, 135, 136, 233, 234, 238, 239, 241, : Please use “Origanum majorana” Ä°nstead of “Origanum majorana”.
  • L26, 125: Please use “Origanum majorana leaf (OML)” instead of “Origanum majorana leaf”.
  • L28, 32, 36, 37, 38, 39, 128, 129, 197, 198, 201, 206, 211, 214, 215, 226, 271, 284, 287, 288, 296, 297, 307, 309, 310, 316, 317, 323, 324, 331, 333: “OML” instead of “Origanum majorana leaf”.
  • L459, L466: “Origanum majorana” should be italicized.
  • L3, 18, 27, 38, 41, 42, 45, 59, 61, 62, 66, 111, 113, 173, 213, 216, 218, 222, 223, 227, 238, 239, 244, 246, 248, 251, 252, 256, 257, 258, 259, 261, 263, 329, 332, 335, 336, 342, : Please use “myogenin” instead of “Myogenin”.
  • L63: Please remove “The”.
  • L28: The explanation of some parameters is open-ended. It is important to state clearly which parameters are to be used.
  • L32-36: They should be rewritten in accordance with their significance level.
  • L37-43: There are repeated sentences that refer to loin, leg, and shoulder muscles. They would be better if they were combined into a single sentence. It is not necessary to write 4% Origanum majorana leaf and % Origanum majorana leaf.
  • L45: The keywords should be sorted alphabetically.
  • L72: It appears that a verb has been forgotten. Please add “found” between “in” and “various”.
  • L80: “Lamiaceae” should be italicized.
  • L81: Please use “region” instead of “countries”.
  • L84: Please remove “of its species”.
  • L86: It is advisable to use the symbol for gamma terpinene, P-cymene.
  • L87, L89: Please use “sweet marjoram” instead of “marjoram”.
  • L89-91: These two sentences should be combined.
  • L98: “…… 144 and 288 ppm oregano esseantial oils….”. Please remove …”essential levels of…”.
  • L100: “Glycyrrhiza glabra” should be italicized.
  • L109: “DLK1” and “IGF1” should be italicized.
  • L126: Please use “crude protein content” instead of “protein”.
  • L134: “Dry Matter basis”: The information is included in the table, so there is no need to include it in the header.
  • L140-142: The text in L140-142 should be aligned left. In L141-142, the trace-mineralized salt content is given; however, the traces need to either be abbreviated or written clearly.
  • L173-177: It looks so complicated. Please use “5'- AATGAAGCCTTCGAGGCCC -3' and reverse 5'- CGCTCTATGTACTGGATGGCG -3' [melting temperature (Tm): 57°C and product size: 100 bp] and for the GAPDH reference gene two primers; forward 5'- ACCACTTTGGCATCGTGGAG -3' and reverse 5'- GGGCCATCCACAGTCTTCTG -3' [Tm: 57°C and product size: 76 bp] were used”.
  • L177, 178, 179, 182, 225, 330: Please use “real-time” instead of “Real-Time”.
  • L174, 177, 182, 183: Please give a space between “…” and “℃”.
  • L188: The font needs to be checked
  • L189-190: Etarget and Eref should be written with subscripts. There should be an explanation of CTGADPH and CTMYOG.
  • L234: Level of Origanum majorana leaf (%)
  • L234, L239: It is recommended that Table 2and Table 3 be revised as “MOL0” and “MOL4”.
  • L245, 258, 260: Please use “goose” instead of “Goose”.
  • L250: The names of muscles should be italicized.
  • L254: Please use “skelatal” instead of “Skeletal”.
  • The names of “MYF5” and “MYOD” should be italicized.
  • L281: Please use “intake” instead of “consuming”.
  • L282: Please use “of” instead of “in”.
  • L291: Please use “because” instead of “Because”.
  • L291-292: It is necessary to revise this sentence.
  • L293, 294: Please use “VFAs” instead of “volatile fatty acid”. In L97, this was previously abbreviated.
  • L301-303: It is necessary to revise this sentence.
  • L296, 299, 301, 303, 305: Please use “FCR” instead of “conversion ratio”. In L99, this was previously abbreviated.
  • L321: Please use “citrus” instead of “Citrus”.
  • L323-324: Please use “had higher lean to bone ratio” instead of “had lean to bone ratio”.
  • L349: “visualization, M.R.M." should be removed since there are no illustrations.

Best Regards

Reviewer 2 Report

Dear authors, it seems to me that this manuscript has great relevance in the scientific world. However, many points affect the quality of the manuscript.

General comments:

Abstract: The abstract is not complete. There is a lack of information of your results. Remember, the abstract is the document that first represents your manuscript. Try to add numbers. E.g.: T4 concentration increased body weight by 5% compared to the control diet. The conclusion does not support the objective.

Material and methods: Taking into account the low number of animals. Please perform the test power analysis to determine that the results obtained are reliable and do not have methodological and/or statistical errors.

Results: Improve the writing style. You need to show more in the results topic. E.g.: … increased by 5% body weight when animals were fed diets containing 40 g of origanum. Furthermore, add the p-values.

Discussion: The current topic of discussion is more of a review or a comparison with other studies. You need to explain what happens. I read few lines about explanations but very generic, like: "the results are probably the effect of the modification of the ruminal fermentation". You have to detail more. How the fermentation was modified? What nutrients reach the intestine? How nutrients are absorbed and what is their metabolism? How the metabolism of these nutrients affects the expression of DNA? etc.

Conclusion: The conclusion has part of the goal; however, the added comment on lines 338-340 of the conclusion diminishes its quality. Try to rewrite the conclusion by focusing on your paper first and add another subtopic like: implications; Please describe all your comments here.

Specific comments:

Line 18: Change “On” instead of “one”.

Lines 37-43: Repetitive. Rewrite it.

Line 40: Change “improve” instead of “better”.

Lines 82 and 87: Please, homogenize the use of the name: Sweet marjoram, marjoram, majorana or Origanum majorana here and throughout the text.

Line 109: Those genes (DLK1 and IGF1) have a name, add them.

Line 110-113: This part is confusing and weak. Rewrite it by describing a hypothesis and a specific objective.

Line 117: This is a low number of animals, add or describe the use of the power test of the current study.

Table 1: Dear authors, are you sure that oregano replaced ground corn in the diet?

Table 1: If oregano replaced ground corn, how is the metabolizable energy the same between treatments?

Lines 147-148: How much blood was collected? On what days was blood collected?

Line 150: What was stored? Blood, serum, plasma, etc?

Lines 150-154: How much sample was used to each analysis?

Line 157: What is “DC”.

Lines 154-163: At what time were meat samples collected for myogenin analysis? How much samples were collected for myogenin analysis? How were those samples stored and treated for myogenin analysis?

Line 166: How were the samples treated before DNA evaluation?

Line 194: Homogeneity was evaluated or only normality?

Line 208-209: Change “hot” instead of “warm”. Here and through the text.

Line 210: Does lean meat refer to a boneless meat? Please describe this in the material and methods topic.

Line 210: What is “cycle threshold”? Please describe this in the material and methods topic.

Lines 222-223: Repeated text on lines 213-214.

Lines 223-224: Why is it high? What values are the references?

Table 2: Are the hot and cold carcass weight measurements correct?

Table 2: Is the Empty body weight measurement correct?

Lines 288-291: Ok, and?

Author Response

(The authors gave the same response as above.)

Reviewer 3 Report

Dear Editor,

Despite the fact that the study was well planned, designed, and implemented, the writing process showed a number of serious deficiencies. In particular, the section in which the results are presented is in need of serious revision. There is a similar problem with the material method section, as it includes some analyses that were not conducted. Other analyses are not presented in a specific order and are not cited. It needs some improvements to the results and discussion sections, but if the conditions listed below are met, then it should be one of the good works in its field that attracts attention.

Major comments:

Since the text contains 50 Origanum majoranas, it becomes monotonous. Therefore, I recommend that you use abbreviations.

L15-L22: It appears that the simple summary section has been abandoned. Although the simple summary contains a good introduction, no statements are provided that reflect the results of this study. Therefore, it would be helpful to modify the sentence between lines 20-22 so that the results of the study are reflected.

In accordance with Instructions for Authors, references should be numbered in chronological order, with a number or numbers in square brackets, such as [1] or [2,3], or [4–6]. It is therefore necessary to correct all references cited in the text.

L73-76: Could you please provide another explanation. This sentence disrupts the flow of the text.

L112-113: As some factors other than gene expression must also be taken into account, this section should be written more emphatically.

L127: The NDF and ADF are not found by calculation. Therefore, this statement should be corrected. Please use "neutral detergent fiber (NDF) and acid detergent fiber (ADF)” instead of “NDF and ADF”, and emphasize that they do not contain ash.

As part of section 2.1, please detail how organic matter, dry matter, ether extract, metabolizable energy are made or calculated.

L128-132: Please write this part in a more clear and specific order. The purpose of using “10% refusals” is unclear. Do you refer to the part of the feed collected for analysis or to the 10% excess of feed that animals can consume by live weight?

For clarity and to avoid repetition, I suggest that the diet that contains 0% Origanum majorana L. and  4% Origanum majorana L. be referred to as "MOL0" and "MOL4", respectively instead of the control and treatment diets.

In Table 1: Please use “Diets*”, “MOL0” and “MOL4” instead of “Used diets*”, “Control diet” and “Treatment diet. Their descriptions should also be corrected in the footnotes. According to the chemical composition, please provide the ingredients of the diets as grams per kilogram of dry matter.

L150-154: The following sentence should be removed from the text. There is no table in the text that provides information regarding cholesterol, triglycerides, albumin, total protein, and serum glucose or testosterone levels. It is important, however, that if the analysis has been performed, the results are presented in a table and discussed appropriately.

L155-160: References should be provided where appropriate.

L195: The reference section should be updated to include " REST, 2009"

There should also be a description of the other statistical model used for the performance and carcass parameters in section 2.3.

L206-219: There is a need to revamp the “Results” section, as it does not discuss the performance and carcass parameters. Each parameter examined should be included in the “Results” section.

L221-241: Tables presented in the discussion section should be transferred to the results section.

L288-291: The results of these studies should be stated.

L296: Please use the word “lambs” instead of “animals” where appropriate. All of the text should be considered in this manner.

L312-315: If it was possible to provide references to which supplements were used in which study, it would be better.

L288, 301, 319, 326: A sentence that begins with "in this regard" does not flow. Similar results can be achieved by substituting "in addition", "likewise", "furthermore", "in the same way" or other pertinent words in place of this phrase.

L329-330: There was no correlation analysis made between myogenin and working traits in the current study. Therefore, this expression may not be appropriate.

L330-335: Since L330-335 are similar and complementary, they are better combined and presented as one sentence.

L329-343: It is important to mention not only the myogenin but also other remarkable results in the conclusion section.

L446-447: It is recommended that this reference be checked.

Minor comments:

L3, 45, 80, 135, 136, 233, 234, 238, 239, 241,  : Please use “Origanum majorana L.” Ä°nstead of “Origanum majorana”.

L26, 125: Please use “Origanum majorana L. leaf (OML)” instead of “Origanum majorana leaf”.

L28, 32, 36, 37, 38, 39, 128, 129, 197, 198, 201, 206, 211, 214, 215, 226, 271, 284, 287, 288, 296, 297, 307, 309, 310, 316, 317, 323, 324, 331, 333: “OML” instead of “Origanum majorana leaf”.

L459, L466: “Origanum majorana” should be italicized.

L3, 18, 27, 38, 41, 42,  45, 59, 61, 62, 66, 111, 113, 173, 213, 216, 218, 222, 223, 227, 238, 239, 244, 246, 248, 251, 252, 256, 257, 258, 259, 261, 263, 329, 332, 335, 336, 342, : Please use “myogenin” instead of “Myogenin”.

L63: Please remove “The”.

L28: The explanation of some parameters is open-ended. It is important to state clearly which parameters are to be used.

L32-36: They should be rewritten in accordance with their significance level.

L37-43:  There are repeated sentences that refer to loin, leg, and shoulder muscles. They would be better if they were combined into a single sentence. It is not necessary to write 4% Origanum majorana leaf and % Origanum majorana leaf.

L45: The keywords should be sorted alphabetically.

L72: It appears that a verb has been forgotten. Please add “found” between “in” and “various”.

L80: “Lamiaceae” should be italicized.

L81: Please use “region” instead of “countries”.

L84: Please remove “of its species”.

L86: It is advisable to use the symbol for gamma terpinene, P-cymene.

L87, L89: Please use “sweet marjoram” instead of “marjoram”.

L89-91: These two sentences should be combined.

L98: “…… 144 and 288 ppm oregano esseantial oils….”. Please remove …”essential levels of…”.

L100: “Glycyrrhiza glabra” should be italicized.

L109: “DLK1” and “IGF1” should be italicized.

L126: Please use “crude protein content” instead of “protein”.

L134: “Dry Matter basis”: The information is included in the table, so there is no need to include it in the header.

L140-142: The text in L140-142 should be aligned left. In L141-142, the trace-mineralized salt content is given; however, the traces need to either be abbreviated or written clearly.

L173-177: It looks so complicated. Please use “5'- AATGAAGCCTTCGAGGCCC -3' and reverse 5'- CGCTCTATGTACTGGATGGCG -3'  [melting temperature (Tm): 57°C and product size: 100 bp] and for the GAPDH reference gene two primers; forward 5'- ACCACTTTGGCATCGTGGAG -3' and reverse 5'- GGGCCATCCACAGTCTTCTG -3' [Tm: 57°C and product size: 76 bp] were used”.

L177, 178, 179, 182, 225, 330: Please use “real-time” instead of “Real-Time”.

L174, 177, 182, 183: Please give a space between “…” and “℃”.

L188: The font needs to be checked

L189-190: Etarget and Eref should be written with subscripts. There should be an explanation of CTGADPH and CTMYOG.

L234: Level of Origanum majorana leaf (%)

L234, L239: It is recommended that Table 2and Table 3 be revised as “MOL0” and “MOL4”.

L245, 258, 260: Please use “goose” instead of “Goose”.

L250: The names of muscles should be italicized.

L254: Please use “skelatal” instead of “Skeletal”.

L255: The names of “MYF5” and “MYOD” should be italicized.

L281: Please use “intake” instead of “consuming”.

L282: Please use “of” instead of “in”.

L291: Please use “because” instead of “Because”.

L291-292: It is necessary to revise this sentence.

L293, 294: Please use “VFAs” instead of “volatile fatty acid”. In L97, this was previously abbreviated.

L301-303: It is necessary to revise this sentence.

L296, 299, 301, 303, 305: Please use “FCR” instead of “conversion ratio”. In L99, this was previously abbreviated.

L321: Please use “citrus” instead of “Citrus”.

L323-324: Please use “had higher lean to bone ratio” instead of “had lean to bone ratio”.

L349: “visualization, M.R.M." should be removed since there are no illustrations.

Author Response

(The authors gave the same response as above.)

Round 2

Reviewer 1 Report

The manuscript is greatly improved making it easier to understand.

However, some descriptions remain incomprehensible in the number and type of samples analysed. What do you mean by (3 biological replicates and 3 technical replicates for each of the three tissues in each of the 2 groups of 12 lambs, totally 648 (12*2*3*3*3) samples).

Authors should improve this part and then the manuscript could be published.

Kind regards

Author Response

Thank you very much for your attention and comments.

We corrected paper Origanum majorana leaf diet influences myogenin gene expression, performance and carcass characteristics in lambs based on your opinions.

We have addressed all of Reviewer’s comments as follows and in the text, using the
“Track Changes” function. We wish to thank the reviewer for the useful comments, which helped us to improve the quality of our manuscript.

  • The manuscript is greatly improved making it easier to understand. However, some descriptions remain incomprehensible in the number and type of samples analysed. What do you mean by (3 biological replicates and 3 technical replicates for each of the three tissues in each of the 2 groups of 12 lambs, totally 648 (12*2*3*3*3) samples). Authors should improve this part and then the manuscript could be published.

To minimize error, each tissue was sampled three times (3 biological replicates) and real-time PCR was run three times for each sample (3 technical replicates) of the three tissues (loin, leg and shoulder muscle).  Therefore, the total number of samples was equal to 648 (12×2×3×3×3) samples including 2 groups of 12 lambs, 3 tissues, 3 biological replicates and 3 technical replicates.

  • Dear authors, I congratulate you on the work you have done. You made corrections to the manuscript according to the reviewersuggestions. I agree with the changes and believe the manuscript can be published as is.

Answer: Thank you so much for your attention and opinion.

  • I have been carefully reviewed your revised article with the id number "animals-2032427". In my opinion, this revised article incorporates most of the points raised in the original draft. However, a few minor points must still be addressed before the document can be published. I believe your research will fill a significant gap in the literature regarding the role of feed additives on the expression of myogenin genes in different tissues of lamb. Please accept my best wishes for all of the authors who contributed to this wonderful work and best wishes for their future endeavours.
  • The title should contain carcass characteristics based on the study. This is why I suggest you use two different titles. “Origanum majorana leaf diet influences myogenin gene expression, performance, and carcass characteristics in lambs” or “Effect of Origanum majorana leaf diet on myogenin gene expression, performance and carcass characteristics in lambs”

3- L22, 155, 345: Please use “lambs” instead of “sheep”.

4- L30-32: Please use “Final weight, average daily gain, hot and cold carcass weight, feed conversion ratio, empty body weight, hot and cold dressing percentage, weight of shoulder, loin, leg, lean meat and lean/bone ratio were measured.”

5- L36: Please use “The MOL4 diet increased ….” instead of “While, adding MOL4 to diet increased…..”.

6- L37,131,208: Please use “feed convertion ratio” instead of “convertion rate”.

7- L80: Please remove “(TH)”.

8- L96: Please use “feed” instead of “food”.

9- L235,236: Please use “feed convertion ratio” instead of “convertion ratio”.

10- L99: Please use “…. some performance and carcass parameters….” instead of ““…. some performance parameters….”.

11- L117-118: Please use “Kjeldahl Vap50 Gerhardt (Germany) was used to determine the crude protein content of samples according to method 976.05.”

12- L118-120: Please use “To determine ash-free neutral detergent fiber (NDF) and acid detergent  fiber (ADF) Van Soest et al. [30] method was performed.” and removed “NDF and ADF obtained by this method do not contain ash.”

13- L128: Please use “each” instead of “every”.

14- L135: Please remove “(EBW)” and “DC)”.

15- L144: Please remove “MOL0 and MOL4”.

16- L145: Please use “Origanum majorana leaf ” instead of “Origanum majorana leaf (MOL)”.

17- L156: Please specify “three tissue” and use “(12×2×3×3×3)” instead of “(12*2*3*3*3)”.

18- L185: Please italicize “GAPDH”.

19- L205: Please use “MOL” instead of “MOL4”.

20- L213: Please remove “always”.

21- L234, 240: Please use “diets” instead of “MOL4”.

22- L306: Please remove “that consumed MOL4”.

23- L339: Please use “study” instead of “stydy”.

Best Regards

Reviewer 2 Report

Dear authors, I congratulate you on the work you have done.  You made corrections to the manuscript according to the reviewersuggestions.  I agree with the changes and believe the manuscript can be published as is.

Author Response

(The authors gave the same response as above.)

Reviewer 3 Report

I have been carefully reviewed your revised article with the id number "animals-2032427". In my opinion, this revised article incorporates most of the points raised in the original draft. However, a few minor points must still be addressed before the document can be published. I believe your research will fill a significant gap in the literature regarding the role of feed additives on the expression of myogenin genes in different tissues of lamb. Please accept my best wishes for all of the authors who contributed to this wonderful work and best wishes for their future endeavours.

Minor comments:

The title should contain carcass characteristics based on the study. This is why I suggest you use two different titles. 

“Origanum majorana leaf diet influences myogenin gene expression, performance, and carcass characteristics in lambs”

or

“Effect of Origanum majorana leaf diet on myogenin gene expression, performance and carcass characteristics in lambs”

L22, 155, 345: Please use “lambs” instead of “sheep”.

L30-32: Please use “Final weight, average daily gain, hot and cold carcass weight, feed conversion ratio, empty body weight, hot and cold dressing percentage, weight of shoulder, loin, leg, lean meat and lean/bone ratio were measured.”

L36: Please use “The MOL4 diet increased ….” instead of “While, adding MOL4 to diet increased…..”.

L37,131,208: Please use “feed convertion ratio” instead of “convertion rate”.

L80: Please remove “(TH)”.

L96: Please use “feed” instead of “food”.

L235,236: Please use “feed convertion ratio” instead of “convertion ratio”.

L99: Please use “…. some performance and carcass parameters….” instead of ““…. some performance parameters….”.

L117-118: Please use “Kjeldahl Vap50 Gerhardt (Germany) was used to determine the crude protein content of samples according to method 976.05.”

L118-120: Please use “To determine ash-free neutral detergent fiber (NDF) and acid detergent  fiber (ADF) Van Soest et al. [30] method was performed.” and removed “NDF and ADF obtained by this method do not contain ash.”

L128: Please use “each” instead of “every”.

L135: Please remove “(EBW)” and “DC)”.

L144: Please remove “MOL0 and MOL4”.

L145: Please use “Origanum majorana leaf ” instead of “Origanum majorana leaf (MOL)”.

L156: Please specify “three tissue” and use “(12×2×3×3×3)” instead of “(12*2*3*3*3)”.

L185: Please italicize “GAPDH”.

L205: Please use “MOL” instead of “MOL4”.

L213: Please remove “always”.

L234, 240: Please use “diets” instead of “MOL4”.

L306: Please remove “that consumed MOL4”.

L339: Please use “study” instead of “stydy”.

Author Response

(The authors gave the same response as above.)
